# Green Assets of Equines in the European Context of the Ecological Transition of Agriculture

**DOI:** 10.3390/ani10010106

**Published:** 2020-01-08

**Authors:** Agata Rzekęć, Céline Vial, Geneviève Bigot

**Affiliations:** 1Research Unit MOISA (Marchés, Organisations, Instituts et Stratégies d’acteurs)-French National Research Institute for Agriculture, Food and Environment (INRAE), CIHEAM-IAMM, CIRAD, Montpellier Supagro, Univ Montpellier, 34060 Montpellier, France; 2Pôle Développement, Innovation, Recherche-French Institute for Horse and Horse Riding (Ifce), 61310 Exmes, France; 3Université Clermont Auvergne, AgroParisTech, French National Research Institute for Agriculture, Food and Environment (INRAE), VetAgro Sup, Research Unit Territoires, 63000 Clermont-Ferrand, France

**Keywords:** equine, horse, environment, green assets, land use, equine grazing, domestic biodiversity, equine and equestrian tourism, equine work, multifunctional review

## Abstract

**Simple Summary:**

Equines have a peculiar place in our society. From livestock to sport, through to landscape managers and leisure partners, equines show a wide range of little-known environmental advantages and assets. Today’s wake-up calls about the environment are progressively putting pressure on stakeholders of the agricultural sector, including the equine industry. This study focusses on the main environmental consequences of equine use and possession in Europe based on scientific and technical sources under the lens of five leading sectors where equines show unique impacts as green assets. Now, more than ever before, it is important to highlight the role of equines as a green alternative in political debates and management practices to give them the place equines deserve in the ecological transition of agriculture.

**Abstract:**

Despite the decline of equine populations in the middle of the 20th century, the European horse industry is growing again thanks to economic alternatives found in the diversification of the uses of equines (sports, racing, leisure, etc.). Equines have many environmental advantages, but the fragmentation of the sector and the lack of synthetic knowledge about their environmental impacts do not enable the promotion of these assets and their effective inclusion in management practices and European policies. To highlight the equine environmental impacts, a literature review was carried out to cover the main European stakes. This work led to the identification of five “green assets”, fields where equines show unique environmental advantages compared to other agricultural productions. These green assets are linked to the nature of equines (grazing and domestic biodiversity), to their geographical distribution (land use), and to their use by human beings (tourism and work). Today, when searching for sustainable solutions to modern environmental issues, the use of equines is a neglected green alternative. Better knowledge and use of equine green assets could partly respond to more ecological agricultural needs and contribute to the development of this animal industry, which has a place in regional development and in Europe’s sustainable transition.

## 1. Introduction

In the European Union (EU), after World War II, equine numbers declined drastically because of the motorization of transport (estimates generally agree that horse numbers decreased approximately 90% in Europe by the 1950s [1]). For example, in France, the total number of equines was evaluated to be three million at the beginning of the 20th century but was less than half a million at the end of this century (Figure 1). Before 1950, horses were largely used for agriculture, transportation, and the army. This was particularly the case for heavy (or draft horses) (represented by the light color in Figure 1), but also for saddle horses (represented by the dark color in Figure 1). The European community was built after the Second World War to maintain peace and ensure the autonomy of its inhabitants regarding basic necessities, particularly food products. To this end, a Common Agricultural Policy (CAP) (which still exists today) was created to improve agricultural production, first to improve cereal yields and then to improve animal productions. However, equines were not included in the CAP’s plans. In this post-war period, the market situation was geared toward productivism, where the disinterest in equines as a source of power and their absence in development policies led the European equine population to collapse. In France, the decline of heavy horses led to the construction of national programs in 1980 to develop meat production [2,3]. After 1970, saddle horses began to be used for other purposes (sport and leisure), which explains the progressive increase in their numbers (Figure 1). This trend was similar in Sweden. In 1920, there were 700,000 horses; then, their numbers decreased to around 95,500 in 1980 before increasing again [4]. However, today, the equine population is still lower (48%) than it was in 1920 (362,700 heads in 2010) [5]. However, the recent increases in equine livestock have not been observed in Mediterranean countries: 87% of horses were lost in Greece between 1983 and 2000, and 31% and 36% were lost in Spain and Portugal, respectively, between 1987 and 2000 [6]. This population decline could be linked to difficulties in national economies during this period.

Today, Europe has 88.4 million cattle, 150 million pigs, 86.8 million sheep, and 12.7 million goats to ensure the animal protein needs of the European population (Eurostat, 2017), but Europe has only six million equines [7], according to the European Horse Network (a non-profit network of stakeholders acting at the world, European, national, or regional level within the European equine sector).

Since the end of the 20th century, the equine industry has undergone significant evolutions linked to the diversification of equine uses: first, in terms of sports, racing, and leisure; and second, in terms of meat and milk productions, traction, therapy, and even companionship. This gave rise to debates on the status of equines (as a farm animal or pet) between European countries but also inside each country. For example, in the United Kingdom, horses are seen as companion animals, whereas in other European countries—France, Germany, or Sweden for example—equines are considered livestock [9].

These issues are particularly problematic because the European agricultural census could be a powerful tool to quantify the equine population at the European level, but today, this census underestimates the equine population because only equines kept by farmers are counted (EU Regulation (EC) No 1166/2008 19.11.2008), whereas, for example, in France, only half of the equine livestock was kept on farms in 2010 [10]. This problem is similar in other European countries (in Germany, for example) [11]. No other database exists at this scale, so the current official figures are misleading. To counter this, the European Union required the creation of a central national database for each Member State that would identify all equines. This requirement was presented in the EU Regulation (EC) No 262/2015 3.03.2015, and the creation of these databases remains a work in progress in some member states today [9]. This lack of data concerning equine numbers in Europe complicates descriptions about their importance and impacts.

At present, the diversity of equine uses leads to a large variety of impacts on the environment, especially when activities are exclusive to equines, such as sports or racing. This creates difficulties in listing and evaluating the environmental impacts of the entire equine industry. However, in today’s European context, local authorities aim to maintain rural activities and to support agriculture in its sustainable transition. These terms, in the EU context, include all policies that search for the transformation of current societal systems to minimize negative effects on the environment and promote innovative projects [12]. A variety of new uses of equines could meet these challenges. In particular, the European Horse Network has expressed the need for a foundation of scientific resources to build arguments and promote equines in European policies and debates. Consequently, this study describes equines not only as animal producers, but also as ecosystem service providers, especially for land use and biodiversity conservation. This choice consolidates the fact that equines are not only seen as a source of agricultural goods (leisure diversification or meat production, for example), but generate a wide range of other externalities.

The aim of this review is to highlight the most important services provided by equines for the environment at the European level. In order to answer public and professional stakeholders’ questions about the inclusion of equines in public policies according to priority, we first met with stakeholders to understand their main issues. Then, a literature review of the available knowledge on the green assets of equines was conducted. Consequently, this paper examines the environmental assets (and limits) of equines that appear to be most important in the context of European policies about agriculture and rural development.

## 2. Choice and Definition of the Main Green Assets

To achieve this goal, the reflection about equine environmental assets started with fifteen interviews among key stakeholders coming from various institutes at the French and European level: the European Commission, the European Parliament, the French Ministry of Agriculture, the International Federation of Equestrian Tourism, the French Permanent Representation in Brussels, and the European Federation of Working Horses. The aims of the interviews were also to estimate the main issues of the equine industry toward environmental challenges. The interviews were semi-directed and were conducted from May 2019 to July 2019. The choice of respondents was made in order to have a large panel of expertise scales (France or Europe) and professional functions (researchers, institutional stakeholders, and professional representatives). The distribution of these 15 interviews was as equal as possible in function and level of decision (Table 1). The high number of interviews from European institutional stakeholders was undertaken to have a better view of the diversity of the issues at this level. Put another way, it was not easy to find professional representatives concerned with the green assets of equines at the European level.

These interviews entailed a reflection based on sectors where equines have particular and specific impacts on the environment, here called “green assets”.

Globally, the most cited impact is grazing, which was described positively, e.g., in terms of pasture maintenance, complementarity with other livestock, and as a carbon sink, even if some respondents insisted on the destructive nature of equine grazing when mismanaged. Equine work and tourism were seen as green alternatives even if the dynamism of equine work was perceived quite negatively, at least in Western Europe. Another point was the key role of buildings and infrastructures related to equines. These buildings are, in most cases, made from wood, which is a renewable material with positive technical characteristics, e.g., isolation, and favorable effects on the landscape. Manure methanization was also mentioned. Indeed, recycling manure by producing energy is a promising way to improve the environmental image of the equine industry. More generally, according to the respondents, equines have quite a positive impact on the environment since they do not eject as much methane as cattle do, they graze, there is a very rich domestic biodiversity that permits a wide range of uses, even as alternatives for engines in agriculture, for example. Often, no negative effects were spontaneously mentioned by stakeholders when asked about environmental impacts.

This process led to an evaluation of the whole European equine industry that considered the current European context and challenges. Green assets were directly or indirectly linked to European rural policies: the maintenance of open areas through grazing, agritourism, and the maintenance of endangered breeds [7].

Green assets are generally directly linked to:The inherent nature of equines (as non-ruminant herbivores whose species presents a high biodiversity of breeds).The geographical repartition of equines and their particular land use.Their use by humans, which can offer environmental benefits. Even if all human activities with equines benefit from a green image linked to the use of an animal, only some of these uses have positive environmental impacts. Of course, the impacts of equine breeding, whatever the equine’s future use, are considered when examining green assets for land use, breed biodiversity, and grazing. We decided to focus on two equine uses that generate specific environmental advantages: equine work and tourism.

This initial information was completed through a literature review of the newest sources possible to build the first state-of-the-art overview of this subject. This review mainly focused on Europe, even though some sources also concerned other continents. In addition to international references, many countries, especially in Europe, developed studies published only in their national language. Consequently, this study was supplemented by French-language references. From March 2019 until November 2019, we used two main databases, Web of Science and Google Scholar to search for articles and reports. To select documents, we searched for the terms “equid*”, ”equine*,” “environment*,” “horse*,” and all words related to green assets, in both French and English.

This literature review highlighted key arguments for five green assets:Equine grazing: Equine grazing incidence is unique because of this animal’s morphology and its physiology specificities, especially regarding ruminants (the main herbivores raised on grasslands). In particular, equine grazing is done on different patches made of lawns and high grasses.Domestic biodiversity: Human and environmental selection has led to a rich diversity of equine breeds all over the world. Some of these breeds are currently endangered and their conservation is an important issue, which could be introduced in European policies.Land use: Equines are present in various areas, especially where other livestock is presently absent. This land use is directly linked to the place of equines in society; as livestock, it is possible to find equines in farms and large areas, but as family pets, equines can be encountered near houses, sometimes on small plots of lands that are not usable for agriculture.Tourism: Equines can be used as a means of transport but also as travel companions to discover wild countries and landscapes.Equine work: Equines are also used in tourism, cities, and agriculture as a source of energy, whereas other livestock are not, at least in Europe.

## 3. State of the Current Scientific Knowledge Concerning the Five Equine Green Assets

### 3.1. How does the Inherent Nature of Equines Impact the Environment?

Equines are non-ruminant herbivores. Equine grazing impacts pastures differently than cattle, sheep, or goat grazing thanks to the particular physiology and morphology of equines, who have a double row of incisors and a high capacity for ingestion linked to the absence of a rumen. They adapt their diet easily according to the available forage. Their behavior also differs from that of cattle in their feed preferences and greater movement when grazing (as there is no rumination rest). These differences induce various impacts on grasslands according to the whether horses graze alone or are associated with ruminants.

#### 3.1.1. Equine Grazing

##### Grasslands in Europe and Their Maintenance

Meadows are lands covered by grasses and legumes and mainly aim to be a feed source for livestock through grazing or mowing. Meadows are known to be a carbon sink: they can stock 60 to 70 tons of carbon per hectare in temperate areas [13,14]. Moreover, the presence of legumes allows for the fixation of nitrogen from the air to the ground. There are different types of meadows: permanent grasslands (retained over 5 years, with possible reseeding after 5 years); semi-natural grasslands (a particular kind of permanent grassland because they exist for more than five years and are known to be among the most species-rich habitats in Europe [15]); cultivated grasslands (seeded each year (or more often than 5 years), and they are the most commonly used in current breeding systems [15]); and other less productive grasslands present in arid or rugged areas, which are called rangelands (shrub lands, steppes, alpine communities, marshes, tundras, etc.) [16]. In Europe, grasslands cover 21% of the agricultural land in the European Union, while croplands cover 22% and woodlands cover 38% (Eurostat, 2015). However, there is a decrease in permanent grasslands in Europe [15], while pastures are known to have different positive impacts: they are seen as pleasant and aesthetic [16] and provide multifunctional goods [15] that produce agricultural commodities and maintain biodiversity, soil, and water quality, even in suburban areas [17], in addition to meeting the needs of herbivorous productions. Animal grazing presents three general consequences: (1) it maintains a certain level of vegetal biomass [18] with the control of invasive species (through the intake of plants [19] and trampling [19]); (2) it has effects on plant metabolism (defence, resistance, and avoidance) [19]; and (3) it enables the creation of ecological niches [20].

##### Equine Grazing Specificities

Equines show specificities that are morphological, physiological, and behavioral. First, equines have two rows of incisors that allow them to graze on short grasses [16] lower than cattle are able to and reach young plants that are easily digestible [20]. This leads to a panel of feed preferences. Equines are graminoids feeders [16] and make less use of forbs and legumes than cattle [15]. Equines’ preferences change when there is feed shortage. Equines move more easily toward less palatable grasses species than cattle [21], particularly in winter [16]. This may be explained by their physiology. The absence of rumination permits more time for feed intake [22]; less methane emissions compared to cattle (100 kg/CH_4_/dairy cow in Western Europe [23], more specifically: 117.9 kg CH_4_/dairy cow/year in France [24], and 18.0 kg CH_4_/horse/year in Western Europe [23] and 20.7 kg CH_4_/horse/year in France [24]); and no limitation of intake capacity due to rumen volume [25]. Consequently, the global intake capacity of equines can be bigger than that of cattle [16] as equines graze longer than cattle [26]. Most equines (except high-performance horses like racehorses) do not need high-quality feed; instead, they are able to live on low-nutrition feed [27]. Moreover, their physiology explains their adaptation to a low starch diet because of the low activity of amylase [15]. They also seem not to be able to digest the secondary metabolites of some plants (such as shrubs) [26]. Because of this, there should be a lower control of shrubs when equines graze them compared to other herbivorous species. However, in extensive Mediterranean conditions with a low stocking rate, shrubs are controlled by local equines [16]. Equines do not appear to be affected by shrubs’ defences, such as thorns [26].

As a consequence of these specificities, equines can adapt their intake through a reduction of feed resources during harsh environmental periods [16]. Some primitive equine breeds are better able to mobilize the bodily reserves they gain during summer to survive during the winter if grass is not sufficient [16] than some specialized equine breeds. For example, in Iceland, despite the cold climate, equine grazing occurs even during winter and is supplemented with the hay refused by cattle or sheep [16]. Likewise, equines can be raised in Camargue, where plants are halophilic and resources are scarce [16].

Equines impact pasture differently than other herbivores because their grazing behavior induces a particular heterogeneity of plant cover [28]. Some areas with high plants are not grazed by equines and could be used as latrines [15], but they could be grazed if there were a lack of resources [27]. Nevertheless, there is also the risk of overgrazing and soil erosion in equine pastures [15]. Equines can impact a pasture, for example, by trampling during periods of exercise [29] or if they are concentrated in small areas. Finally, gnawing on trees in semi-natural pastures was reported [20] due to the lack of minerals [15], but this can be reduced with supplementation [30].

Studies on exclusive equine grazing show the positive impacts on flora, including an increase of legumes in France [20], a control of competitive grass [15,20], and control of some shrub species in Camargue [20] (e.g., Vaccinium *myrtillus* by trampling [16,31]), alongside negative impacts, such as the limited control of fast forest regeneration in boreal conditions [15] and an increase of foams in Iceland [20]. In particularly harsh conditions, some lands are maintained by equines to decrease fire risks, for example, by reducing the aerial biomass of gorse in Galicia [16]). Equines are able to preserve and maintain pastoral biodiversity [15] by grazing in areas abandoned by agriculture. This process is identified as a specific threat to habitats and species by the European Union, as the invasion of some plants left non-grazed by livestock leads to landscape closure [25].

Equine grazing also impacts fauna:The populations of small herbivores increased due to high-quality vegetative regrowth.Insectivorous birds, such as the spoonbill, appeared in pastures grazed by equines in the Netherlands (as with ducks in French wetlands) [20].The wolf population in Galicia, Spain, was maintained partly thanks to ponies bred in semi-natural conditions. These one were the preferential feed source of wolves [27], which consequently did not attack other livestock. As a result, farmers felt less disposed to shoot them down. Finally, ponies were an indirect way to conserve the wolf population.

Pastures can be grazed by equines only, by two or more herbivorous species, or mowed. All these types of pasture management methods allow animals to feed and exert their own effects on the environment (biomass, grassland evenness, and effects on butterflies) depending on the area (grazing seems to be a better option in Central Europe and mowing may be more suitable in Southern Europe) [32]. In boreal conditions, continuous equine grazing seems to be less beneficial to biodiversity than alternative grazing regimes (late grazing, years without grazing, and mowing) [15]. In Sweden, year-round equine grazing increased the pasture quality and diversity compared to mowing [28].

##### The Impact of Cattle and Horse Mixed Grazing on Meadows

The heterogeneity of plant cover may be beneficial for flora and fauna but can lead to pasture destruction when it is mismanaged. Indeed, non-grazed areas can be invaded by shrubs, while grazed areas can be overgrazed. To prevent this situation, mixed grazing with two herbivorous species could be a solution. Mixed grazing is “a practice of stocking two or more species of grazing or browsing animals on the same land unit, not necessarily at the same time but within the same grazing season” [16]. Cattle and equines graze preferentially in similar habitats, such as grasslands [26]. This can lead to a competition for resources but also complementarity because of their different foraging behaviors. A survey carried out on farmers who raised both (dairy and/or beef) cattle and horses in the French central mountains highlighted the equines’ ability to exploit the grasses refused by cattle (total surveyed farmers: 25) [33]. In Massif Central (French mountains), when equines were introduced on cattle pastures, their nutritive value increased because of the development of higher nutritive plants (within the following conditions: 350–440 kg live weight per hectare and as many horses as cattle) [16,20]. When bred with cattle, equines could graze in poorer pastures not useable by cattle because of their low nutritive requirements [33]. Moreover, in winter, equines can remain on pastures and continue maintaining them while cattle are held in stables [16]. However, mixing cattle and equine grazing can generate problems, especially when increasing the number of animals in the pasture. For example, when the equine stocking rate is too high, cattle may be disadvantaged because they are unable to eat enough [27,34]. The main consequences of mixed grazing on grassland management are:A lower workload: Because their grazing behavior is different from and complementary to cattle, equine replace the use of machines for mowing grass refused by cattle and for crushing wastelands [22,33]. In the absence of equines, farmers would need to use the roller chopper more frequently [22].A decrease of the parasitic burden: This may be explained by different host sensitivities and improved nutritional status (intake of various plant species) [16,35].A better control of woody species: In Massif Central, woody species were better controlled when pastures were grazed by cattle and horses than by cattle only [20].

In France, equines are often raised with cattle in grassland areas [10]. A study of 51 farms located in the highlands, breeding cattle (for beef or milk production) and heavy horses together, highlighted the common practices and confirmed the preceding results about the advantages of mixed grazing [22]:Equine grazed mostly after dairy cattle but grazed simultaneously with suckling cows and heifers.Equine grazing helped to remove grasses refused by milked cows.Equine were present on small plots, fields far from stables, and on poor pastures.

In this survey, equines comprised on average about 10% of the total livestock in terms of livestock units. Heavy horses were bred for meat production mainly. This breeding process is seen as a complement to the production of cattle for use in grasslands but is not considered a significant source of income. On the other hand, thanks to equine pasture management, the mechanical maintenance of meadows decreased, which can be seen as an indirect contribution to the greater efficiency of forage systems in the highlands.

Beyond grazing specificities, the inherent nature of equines leads to another environmental asset: equine species present high biodiversity among their breeds, which is shaped by the environment and human beings.

#### 3.1.2. Domestic Biodiversity

Several equine breeds are well adapted to poor grasslands and semi-wild breeding systems [18,26,27] because their format and size seem to ease their growth in these specific areas [36].

For a long time, and still today, natural and human selection have affected breeds, including those that live in semi-natural or particularly harsh conditions [25]. Human beings selected the most suitable breeds for different uses (traction, meat production, racing, and sport), leading to a high variety of sizes, formats, and phenotypes of equines sorted into breeds. A breed may be defined as a breeding pool of individuals that share a common phenotype (which is typically purely morphological, i.e., coat color, height, etc.) [25]. Worldwide, in 2011, there were 397 equine breeds according to the Universal Equine Life Number (the international lifelong identification number for equine, www.ueln.net). Germany (46 breeds), France (37), and the United States (34) were the countries with the greatest numbers of breeds. Of all these breeds, three-quarters were saddle horses (sport, recreation, and ponies), 15% were heavy horses (France had the highest heavy breeds number with 9), and racing breeds represented only 5% of breeds [37]. Today, uses of equines are diversifying, and competition between studbooks is increasing because of the internationalization of the equine industry. Consequently, some equine populations collapse when they are not useful anymore. For example, worldwide, 60 donkey breeds are known, but only 28 have had their morphologies described [38]. Only one donkey breed out of 28 is considered to be not threatened by the Food and Agriculture Organization of the United Nations, as 80% of the donkey population disappeared over 20 years in Europe [ibid.].

Being part of a breed means being identified in a studbook, which is a book of genealogy where breed standards are established. A closed studbook does not allow foals whose parents are not identified in the book; this keeps the breed “pure.” However, to improve the traits of interest, studbooks may be opened to some other breeds that will shape the breed. As an example, the pure-bred Arabian and thoroughbred breeds underlie 25% of the genetic variability of 500,000 saddle horses from 55 different breeds, born between 2002 and 2011 in Europe [39]. A consequence of this intensive direct selection is the specialization of breeds toward modern uses. This specialization is a threat to the versatile, multi-skilled, and generalized breeds. Conversely, these breeds can be seen as an emblem of a region that needs to find new approaches to remain competitive. Finding economically sustainable alternative activities in environmental conservation projects (e.g., animal traction, equestrian tourism, and meat and milk production) would be an interesting way to conserve local and endangered breeds.

European aids for endangered breeds help breeders maintain their activity, even if equine production is no longer economically competitive (Regulation (EU) 2016/1012 of the European Parliament and of the Council of 8 June 2016). This underlines a contradiction in common European programs that give funds based on a small number of animals to preserve animal biodiversity and not just their skills [40]. This kind of breed development is not sufficient to ensure economic viability [41]. However, these programs conserve invaluable environmental services and cultural heritage [25,42].

Equines, because of their essential nature, directly impact the environment. Nevertheless, these impacts may differ depending on the location of the animals.

### 3.2. How Does the Spatial Repartition of Equines Impact Land Use?

Equines require forage areas for grazing and for preserved fodder harvesting. Equine grazing may lead to the maintenance of open areas and a possible improvement in the agronomic quality of grasslands [43]. Breeding farms, riding schools, racecourses, and trails are other kinds of indirect land uses by equines. A French region typology of equine farms highlighted that equines are present in 91% of all cantons (exclusive of Corsica) [44], which means that the equine industry extends to various areas. Equines are also present in suburban areas: 75% of equines are encountered in the most inhabited areas in Sweden, for example [45]. This is also the case in Scotland [46], Germany [47], France [48], and Belgium [17]. Equines are also present in Polish post-agricultural lands and forests [49], Spanish heathlands [27], and British grasslands [25]. Studies across Europe agree that equines, whatever their use, are located in various kinds of lands: (a) suburban areas, (b) rural areas, and (c) sensitive areas, such as mountains [17,44,45,46,47,48].

#### 3.2.1. Suburban Areas

Pastures in suburban areas are seen as an extensive method of farming and a good way to conserve water quality; however, these pastures are disadvantageous for agriculture [30] because of their small available surface, the presence of housing and non-rural neighbors, and land conflicts. Equines are kept in suburban areas thanks to urban demand [47]; equines are not only seen as a source of agricultural income [50] but as a leisure activity [51] or even as a family “member” whose place is near the home [43]. As a link between urbanization and rurality [51], equines are present in transitional areas that have been abandoned by agriculture but have not yet been developed by urbanization [43]. This kind of “sub-agriculture” [43] may be called “soft urbanization” [17] because equines are a spatial and functional link between residential areas and agriculture, and are also of concern in land conflicts with other agricultural productions. In fact, these conflicts for land force equines to reach the edges of urban areas [43]. Nevertheless, equines remaining in these areas mobilize a large array of services, including veterinarians, feed industries, trainers, equipment sellers, and transporters [47].

Some studies quantify the presence of equines in suburban areas, underlying their growing importance. In France, research conducted on 49 municipalities showed that equines use between 1% and 3.5% of suburban areas depending on the region [43]. In Sweden, it was shown that the density of equines can increase up to 6 horses/km² near urban areas but that specialized equine farms near suburban areas keep more equines than farms in rural areas [51].

In terms of environmental impacts, the presence of equines is comparable to the introduction of nature in the city, which improves landscapes [51]. Equines can graze on small plots near forests and gardens (for example, in Flanders (Belgium) and the Netherlands [17]). Their presence yields positive changes, such as positive land management, added landscape value, and job creation [43,45]. Equine grazing is perceived as being a positive element of landscape management in Belgian suburban areas [17] and in Sweden, where the terms used by inhabitants to describe the equine presence were globally positive: ecology, landscape managing, and useful [51]. The presence of equines in these areas opens the possibility to include equines in reflections about urban planning [43,51]. Nevertheless, the high density of equines in small areas could pose a threat to the environment [47] through problems, such as overgrazing, droppings concentration, and destruction of landscapes, because of the creation of mismanaged infrastructures (overgrazed paddocks, and horse-riding rings) [45]. Equines are also a source of odor, insects, and lack of safety [43,45], thereby leading to land conflicts. In Belgium, because of the population density of the country, farmers and equine owners fight for land [52]. Indeed, only 10% of land is considered to be rural (depending on the density of inhabitants) in this country, but agriculture is present in 45% of the total national area, and 1/3 of grasslands are grazed by equines, underlying their high presence in suburban grasslands [17]. The presence of equines in these areas does not always imply grazing because equines can be fed with cereals and preserved forage. In some cases, the available area does not allow grazing, and equines are kept in small unproductive plots or stables [34]. For example, in the Berlin suburban region, there is no grazing in specialized and intensive farms [47]. This raises specific issues related to overgrazing and manure management. More studies about grazing in these specific areas are needed [17].

It is important to note that the presence of equines in suburban areas depends on urban sprawl (horse riding schools near cities), but sometimes urban sprawl depends on the presence of equines (owners who want to live closer to their horses) [43]. This kind of agriculture in suburban areas could respond to sustainable issues according to some authors [47], but communication actions should be deployed in order to raise awareness about the risks of overgrazing, droppings concentration, manure management, safety, and land laws.

#### 3.2.2. Rural Areas

Rural areas are the main production areas for animal feed and for other derived products, such as straw. In France, equines use from 1.5–6% of the total rural areas. Equine numbers depend on the type of agriculture: the numbers are lower if there is more professional agriculture (e.g., if the lands are used for food production) and they are higher if non-professional agriculture prevails (e.g., retired persons or multi-active land owners) [43]. The presence of equines may be combined with other local agricultural production, thereby maintaining pastures even in intensive agricultural areas [44]. Farmers can experience benefits from the presence of equines on their land, such as selling equine feed, letting the equines graze on unproductive pastures, or receiving manure.

#### 3.2.3. Sensitive Areas

A sensitive area has a long-term capacity to maintain and enhance natural resources, such as soil and water quality, biodiversity, and the landscape. In such areas, agriculture is more constrained than in the lowlands. Mountains are part of this definition. Other kinds of sensitive areas include natural rangelands in harsh climates and abandoned agriculture areas invaded by shrub, wetlands, or heathlands.

Some sensitive areas produce difficulties when using machines because of the slope or soil depth. Consequently, in these areas, animal husbandry seems to be the most adapted solution to preserve landscapes and maintain economic activity. For example, in French plains, farms breed mostly saddle and race horses with high economic value [10], whereas few heavy horses are associated with cattle farming in the uplands, in an attempt to improve grassland management despite their low economic value [22]. In this study, up to 15% of farmers said that they would abandon some parcels of land if they did not possess equines [22]. In Poland, ponies named Konik Polski use forests and post-agricultural areas, exerting several impacts on them, including a reduction of shrubs and bushes through trampling, an increase in coprophagic insects, an increase in birds (thanks to greater food resource diversity and hiding place availability), and an increase in the interest and awareness of nature and ecology for tourists and local inhabitants [49]. In Galicia, Spain, where transhumance was abandoned, the presence of equines raised in semi-wild conditions helped to restore heathlands [16]. Heathlands and grasslands in the United Kingdom are usually grazed by sheep, but there have also been studies on the reintroduction of ponies to these areas, where the authors recognized the value of equines as conservation grazers [25]. Rewilding areas abandoned by agriculture with large mammalians, such as equines, are a proposed solution to maintain the important functional links between plants and pollinators in grassland ecosystems in Sweden [18]. Some releases of equines may be a threat to ecosystems, as sometimes breeds are not indigenous to the region. Specific attention should be given [25] to wisely choosing breeds that are to be introduced in natural areas, as well as the density of the released equines. In the “Parc des Volcans d’Auvergne” (an environmentally protected area in France), some authors prescribed the use of mixed grazing with equines and cattle to restore the area [53]. Rangelands are also affected by the definition of sensitive areas. Mediterranean zones show the possibility to access a diversity of pastures (salty pastures, marshes, and rice stubbles), which complement each other and permit local equine breeders to lengthen the grazing season [16]. In areas where the soil is fragile, like in low mountainous olive groves, donkeys are usually used to control grass growth because they weigh less than other heavy animals, such as cattle or horses [38].

In highlands, preserving natural biodiversity when maintaining human activity is possible through agriculture, and more specifically, through animal husbandry. Indeed, the maintenance of the open landscape is particularly interesting for its biodiversity, as explained above, to preserve vegetal resources in non-arable lands, but also to manage areas for human activities, such as hiking or skiing. Because of their slopes and peculiar climates, arable lands are scarce, and pasture meadows are favored as a source of food for livestock, thereby spearheading agricultural products, such as meat and milk.

To conclude, equines take part in the problems of land pressure but have complementarities to agriculture and urbanization in terms of their functionality for land use and maintenance [43]. Other environmental advantages of equines come from their use by humans for working or leisure activities.

### 3.3. Animal Uses Serving Environmental Issues

Equines have been used by humans since their domestication. Activities with equines are often seen as “natural.” However, not every use of equines is environmentally virtuous; for example, horses travelling by plane for international competitions or races. According to our findings in the literature review, two uses present interesting environmental assets: equine work and tourism.

#### 3.3.1. Equine Work

In this study, a working equine refers to an equine who is used to work with humans; provides energy that can be substituted by other sources of energy, other machines, or types of transport; and generates earnings [54]. Sport and leisure horses, equines used for therapy, race horses, and equines bred for meat or milk productions are not affected by this definition. There exist four primary non-exclusive types of equine work: agriculture (mostly vineyards and market gardening in organic production systems), forests (logging), human transportation (such as equines drawing a carriage for tourists or schoolchildren), and public service missions (watering, garbage collection, and mounted police).

In the world, there are ten times more animals used as sources of traction energy than motorized tractors. In “developed” countries, 26% of the land area is managed by animal traction (versus 52% in “developing” countries) [54], especially in sensitive areas [55] or in mountains [56], where plot structures (slope and soil quality) make mechanization difficult. However, these numbers have decreased over time in Europe. For example, in Poland, the percentage of horses used in agriculture compared to all sources of energy decreased from 93.8% in 1950 to 1.73% in 2009 [57]. In the context of productivity gains in post-war Europe, it was necessary to work on bigger areas in less time. Machines seem to be more adapted to this aim than animals, as their use increases sowing, treatment, and harvesting speeds, along with work efficiency, as well as decreases the time dedicated to crops. Nevertheless, a full replacement of animals used in traction by machines may be perceived as a heritage loss [56] and a threat to the environment (soil quality, for example).

##### Equines as a Potential Source of Renewable Energy

Equines consume fodder, which is considered a renewable source of energy because it does not involve fossil energy in the narrowest sense (unlike fossil fuels or biofuels) [58]. In fact, biofuels may be considered, in some cases, to be a renewable source of energy, but they need the same arable lands as crops and are the focus of land conflict debates. Grasslands used to feed equines could be located in non-arable parts of the territory, as noted above. A one-day harvest allows for enough forage to feed an equine for one year in Switzerland [54]. A total of 0.6 ha of alfalfa, 0.5 ha of oats, and 0.5 ha of wheat for straw are enough to feed two horses working on 14 hectares for 140 days a year in Croatia; for the remainder of the year, these horses stay in stables, where they are fed the by-products of crops, or graze on roadside vegetation or in orchards [58]. Moreover, grasslands and areas worked with equines can be fertilized with their manure. In the case of biofuels, nitrogen must be imported or manufactured and is spread on plots where it evaporates into the atmosphere, providing the main source for N_2_O emissions [58]. The animals must be fed all year, whereas machines can be used occasionally and refuelled infrequently. Despite this disadvantage, equine work allows farmers to attain better feed and energy autonomy [54], to highlight a traditional vision, to be appreciated by urban inhabitants [58], and to maintain a diversified gene pool through the use of local equine breeds [55]. Finally, equine work is considered by some authors as a form of sustainable agriculture [58].

##### On Arable Lands

On arable lands, soil compaction is known to be the most severe form of degradation in conventional agriculture [55,58]. There is a difference between the paths made by machines (continuous, because of their tyres, with deep soil compaction) and equines (intermittent, because of their hooves, with superficial soil compaction) [58]. The soil porosity was higher after using donkeys or cattle compared to a motorized machine [55]. A comparison between the use of a donkey and a motorized machine for the ploughing, fertilization, and preparation of rapeseed in the context of the high hills in northern Italy was made thanks to the life-cycle assessment (LCA) approach for which inventory data were taken from the GaBi 4.0 database. All aspects related to the life spans of animals were considered, except the end of life: pregnancy, growing and maintenance (health care, feed, keep, and equipment), and work. For machines, material acquisition, manufacturing, utilization, transport, and disposal were considered. This information was acquired through interviews with animal owners, field measurements, and technical reports from manufacturers. The results showed that, for the same amount of carbon emissions (1 kg eqCO_2_), a donkey was able to prepare 330.63 m² of land, whereas the machine prepared only 18.69 m² (three operational stages were considered on a 1000 m² functional unit: ploughing, application of the fertilizer, and seedbed preparation (harrowing and opening seed furrows). Manure from donkeys was assumed to be applied as a fertilizer, so environmental impacts from fertilizer production were avoided in the case of animal traction) [56]. If the fossil fuel used for machines had been replaced by biofuels, the relative effects on the environment could have been 9% lower. When comparing classic machines and donkeys, these effects were 97% lower [56]. In Ireland, yields were greater when animal traction was chosen after the long-term use of tractors [58]. It is important to note that equine traction is well adapted to small areas. Finally, there are not enough studies about equine work in mountainous areas, where they could have particular assets [55].

##### In Forest Areas

Equines are known to be more drivable than machines in forests, or on rugged or narrow fields. Thanks to this skill, there is less damage to residual trees [55] because machines need more space to access fields [55] and create disturbances [56]. Without counting trail development costs, the use of an equine was more profitable up to 50 m [55]. This distance increased up to 200 m when trail development costs were considered [55]. A comparison between the use of a mule and a motorized machine in a one-kilometer distant forest was made thanks to the LCA approach and showed that, for the same amount of carbon emissions (1 kg eqCO_2_), a mule was able to bring 311.30 kg of wood, whereas the machine brought only 79.64 kg of wood (three stages were considered: (i) felling (individual cutting of trees), limbing (removing branches), and bucking (cutting into logs); (ii) yarding (collection of logs); and (iii) transport from the forest to the farm (1 km). The functional unit was set to 100 kg of wood at the warehouse. In the animal traction scenario, a mule was used in stages (ii) and (iii)) [56]. If the fossil fuel used for the machines was replaced by biofuels, the relative effects on the environment could have been 26% lower. When comparing a classic machine and a mule, these effects were 74% lower [56].

##### Other Agricultural and Territorial Works

Equines can also be used in:Old vineyards, because their drivability permits work in narrow rows and on terraced or steeped fields (Douro River Valley, Portugal; Bordeaux, France; Sibeira Sacra region, Spain [55]).Greenhouses, because their drivability allows for precise work [54] and can be highlighted in ecological production.Natural areas, where they are less noisy, degrade the soil less, and frighten local fauna less [54], thereby enabling them to work in protected and sensitive areas. It is possible to compare this to the consequences of equestrian tourism on wild fauna, which are perhaps less frightened by equines than by pedestrians or bikers [59]. Mules are still present in some European areas, such as national parks, where it is impossible or forbidden to use motorized tractors [38].Cities where they decrease the carbon footprint and are used as “city pacification” agents [60].

#### 3.3.2. Tourism

According to the International Federation of Equestrian Tourism (FITE), the term “equestrian tourism,” which emerged in the 1950s [61], concerns all outdoor activities with equines outside of residential areas. Indeed, it is necessary to distinguish between equestrian tourism and equine tourism:Equestrian tourism comprises itinerant journeys with a ridden or hitched equine or on foot supported by a pack equine.Equine tourism concerns all activities devoted to equines, in their presence or not, that attract tourists, including sport events, cultural events, races, fairs, museums dedicated to these animals, riding courses, etc.

In addition, the two kinds of tourism linked to equines can be local (i.e., tourists move inside their region of origin) or non-local (inter-regional, international, etc.).

France is the third-largest country in terms of rider numbers in Europe (behind the United Kingdom and Germany), but is considered to be the leader in equestrian tourism and the first travel destination, with 60,000 kilometres of equestrian trails in 2011 [61]. Other countries highlighted in the European report on the equine industry in 2001 are Greece and Portugal, where donkeys are still used for tourism [62]. Equestrian tourism has expanded through farm diversification called agritourism, which is affected by the European development policies for 2014–2020 [36]. This kind of tourism can involve farmers who want to promote local breeds to preserve culture and tradition, as is the case of Camargue horses related to specific bull farming in marshes of the Rhone River estuary in the south of France (Figure 2) [61] or Icelandic ponies in Iceland [63], where tourists want to find a link between nature, animals, and local culture.

##### Impacts of Equestrian Tourism

Equestrian tourism is a form of sustainable leisure [61], though there are very few studies on the direct impacts of using equines in tourism; these studies are mainly American (12 in total, as of 2019) and Australian (six in total, as of 2019) [64]. This is why it is important to contextualize the results presented below, as climate and cultural history are not the same in every region. Europe has always been a host to large mammals, whereas Australia never housed such animals before 1800 [64]. A comparison between hiking, cycling, and horse riding [64] shows that the impacts on the environment were the same between these means of transport but they differed in the degree of impact (e.g., soil compaction and erosion, loss of organic matter biomass, and biodiversity losses). The two most severe impacts of equestrian tourism are: (1) nitrification of rivers and soils because of the overconcentration of phosphor in poor soils, and (2) zoochory through fur and manure, which raises the risk of invasive plants being spread in protected areas [64]. On the other hand, this spread may be beneficial for the flora diversity of poor soils.

Equestrian tourism is also an illustration of soft roaming. Indeed, nature-based recreation activities impact wild fauna (most of them, even if non-motorized, have negative impacts on birds in terms of their metabolisms, behaviors, and habitat disturbances [59]). There are no studies concerning equestrian travel, but it is possible to consider a softer approach for wild fauna; wild animals are, perhaps, less frightened by horses than by pedestrians. In addition, the trails used for horse-riding are a softer way to adapt land to tourism than roads. Moreover, in sensitive and protected areas [61], horseback tourism also creates and maintains trails in a useful state for other users.

The negative impacts of equestrian tourism must also be mentioned. Infrastructures are not always adapted to equestrian tourism, and this shortfall can present destructive impacts on the environment. Indeed, during a long-term journey, a horse must rest every 20–40 kilometres but this is not always accomplished [65]. In Poland, for example, on the longest national equestrian trail (2100 km in 2012), there are only 36 liveries; this means one stable every 41 km [65]. When searching for a campsite or pasture, horse-riders can destroy sensitive or protected areas [64]. Moreover, from surveys carried out on equestrian tourists, security, comfort, and conviviality were more important when travelling than ecosystem conservation [64].

##### Equine Activities and Tourism

Because of the wide definitions of equine tourism, it is difficult to list all ecological effects of such tourism. However, some reflections about this issue can be highlighted. In France, for example, the professional organization of the horse industry provides awards to infrastructures that put efforts toward improving their environmental impacts, such as riding schools in Camargue [61] or the numerous stud farms and boarding stables all over the country, in the form of environmental labels (www.label-equures.com, accessed 27.11.2019). This is also the start for reflecting upon the environmental impacts of equestrian events (the transport of horses and persons, the use of natural resources, etc.) in France through an evaluation of the most impacting positions, like wash areas [66]. These efforts provide an early ecological wake-up call for all events, including equine-related ones, and need further research.

## 4. Discussion

### 4.1. Links between Green Assets and Specific Issues

The green assets highlighted above are not completely independent but are mostly linked to each other.

This fact can be first illustrated through the example of the Camargue horse, which plays a key role in land planning and tourist development. This equine breed is known to be a representative of local cultural heritage (Figure 3). Raised in the wetlands, Camargue horses are robust and largely participate in the maintenance of this sensitive area through shrub grazing. They are also hardy enough to be used to control and move cattle herds in these vast swamps. The Camargue region attracts tourists because of its particular fauna (such as flamingos, horses, bulls, etc.) and flora (halophilic plants). The Camargue equine breed is also well adapted to be ridden by tourists to discover the natural environment of this area. Finally, Camargue horses can also be found in suburban areas, such as in riding schools or pasture areas near houses.

Each green asset has its own specificities and issues but some of these issues are common to several green assets. For example, because of the task intensity, animal traction sometimes must be assisted by another source of energy (in cities for example) [60]. To avoid soil compaction, innovative projects have emerged to improve machines, but the use of equines is not always proposed as a solution [58]. Finally, there is a need to improve the equipment for animal traction [55] and to inform stakeholders about this need. Presently, the use of equine work depends on the geographical and market opportunities created thanks to these animals [56]. These opportunities should also be developed for threatened breeds whose population have collapsed since the middle of the 20th century. Regardless of the species, these communities must face common challenges [41], such as how to ensure their competitiveness on the international market while maintaining the appearance of an iconic breed that is illustrative of regional culture and heritage, and how to conciliate the development programs and conservation programs for an already threatened breed. The uniqueness of some breeds is used to bring tourism into the breeds’ native region, such as Icelandic ponies in Iceland [63], even if tourism may be a source of conflicts with other users and even sometimes a source of the deterioration of protected areas [61]. These conflicts may also appear in land use. Equine owners are often disadvantaged toward more “agricultural” productions [17]. In suburban areas, equines stand on transitional plots, which are rented or borrowed [43], within the urban network. This presence may lead to disturbances, such as smells and noises. However, in Sweden, a survey made in the suburban areas where equines were encountered showed that the inhabitants were less annoyed by equines than by other disturbances, such as noisy roads or mowers [51]. Urban planning does not consider the specificities of the equine industry. Equine owners want to keep their animals close to their houses and urban centers but are restrained because of land conflicts. If urban planning undertook a multifunctional approach for every activity [17], there would be fewer conflicts of interest [51]. Conversely, intensive equine holdings on small plots, as is often done in suburban areas, can be harmful to the landscape. This is why “horsiculture” is sometimes seen as degrading for the environment [45] as it is directly linked to the specificities of equine grazing that leave areas overgrazed if badly managed. In addition, in most cases, only broodmares and growing horses graze, while stallions or equines in training do not [22]. Moreover, the workload increases when horses and donkeys graze if they are trained daily because, sometimes, it is necessary to pick them up from distant plots [22].

Facing this context, each member state of the European Union has specificities concerning equines. Some countries orient their equine industry more toward sports and racing (e.g., England and Germany) and do not consider equines to be a source for rural development [67], whereas others highlight their equine sector through native breeds and tourism, such as Iceland [68]. Thus, impacts on the environment will be as diverse as the uses of equines and farming in each area. In some countries, equine husbandry is perceived as an intensive process (no grazing, many individuals on small plots, main outlet exported, etc.), for example, in Belgium, the Netherlands, or Luxembourg, because of the lack of available land. In other countries, equines are still used as a source of energy or entertainment, with grazing on large lands and sometimes in semi-wild conditions, such as in Poland [49]. In occidental Europe, land use by equines depends mainly on the type of area (rural, suburban, or sensitive), as discussed in the aforementioned studies. This grouping may be different in other countries, such as Romania, where horses are still used in agriculture [69]. Thus, the numbers of equines may still be anecdotal in suburban areas.

Consequently, it is important to determine how to best take advantage of the green assets of equines.

### 4.2. How to Better Use Equine Green Assets

#### 4.2.1. For Equine Keepers

This knowledge is partially taught in agricultural training. Indeed, there is a lack of training courses and technical information for small equine owners to teach the specificities of equines’ relationship to the environment. Difficulties arise from the atomization of horse owners and the diversity of horse keeping. Moreover, owners mostly raise horses in small areas with low productive value, so the technical improvements of pastures are difficult to apply. The risks of injury or escape have also been is raised by equine breeders, even if equine grazing itself is not to be feared. Advice for management practices needs to be adapted to the specific conditions of each horse keeper according to the type of meadow, the herd size, the physiological and nutritional needs of the equines, and welfare requirements. To improve this situation, communication and teaching about pasture management practices, related risks, and threats are necessary. Raising awareness and improving management practices could improve equine environmental impacts, highlight their green assets, and better use these assets in everyday practice. It is already possible to propose some examples of practical recommendations to equine owners:In order to avoid the overconcentration of manure in suburban areas, equine owners may build reliable partnerships with local farmers, who can use manure as fertilizer. Manure can also be recycled and rapidly composted to improve soils in city parks. Another solution is to transport manure to methanation firms for energy production [70].Equine grazing has interesting characteristics in pastures and shows complementarities with other herbivores, such as cattle. Thus, associating these animals could be a first step toward improving pasture quality and maintenance.When searching to buy an equine for leisure or tourism, looking for a native breed could be a good option if the future owner wants a hardy equine that is well adapted to the local climate. These breeds may value local feed resources better and more cheaply than other breeds. At the same time, this act would help to conserve threatened breeds, facilitating a cultural development of the region and maintaining the genetic diversity of equine species.When travelling on horseback, it is important to follow trails that are dedicated to horse riding, to avoid protected areas, and to take care of the paths.Equine work represents a diversification opportunity for riding schools. This diversification can be achieved using equines that are already present in the structures for maintenance tasks, either on site or in collaboration with local municipalities (service provision).

Despite these few examples, it is necessary to develop further recommendations for equine keepers in order to establish clear aims, build a reliable argument, and ensure adequate follow-up. These recommendations could be spread by local authorities, teaching centers, and the professional or public institutes responsible for the horse industry. As a contribution to advancing knowledge, this literature review is a first step that will need regular updates to enhance the advisors’ arguments.

#### 4.2.2. For Institutional and Political Stakeholders

Equines are often forgotten in political debates as they are seen as both livestock and pets. This duality is exacerbated by the large diversity of stakeholders responsible for the equine industry within Europe, including ministries of agriculture, sports, and tourism; technical institutes; research institutes; national and regional associations; breed associations; and equestrian federations. Moreover, there is a lack of practical information and courses on equine green assets that are usable by stakeholders. Promoting equine grazing and communicating about its benefits on the environment can improve different situations. This could also help integrate equines into political debates and help develop research on this topic. Hence, it is possible to propose some practical recommendations to institutional and political stakeholders:Equines are an interesting alternative for the maintenance of small abandoned lands in suburban areas that could be promoted by local authorities.In regions where grasslands or rangelands are important, local development policies could include aids in favor of the equine industry, for example, subsides for cattle farmers to also hold equines, or for the creation of numerous platforms to help horse owners meet farmers for feed purchases, pasture grazing, or the use of manure.Regional subsidies could support the breeding and keeping of local breeds. These could also be integrated in local tourist events or as a vector for job-creation.The trails used for equestrian tourism and camping sites should be framed well to keep tourists from disturbing natural areas. Moreover, every trail should clearly indicate whether it is adapted for equines to ensure that equestrian tourists use the trails safely. Further, linking equestrian tourism stakeholders with stakeholders from the tourism sector or those in charge of protected areas could be an interesting way to develop collaborative actions to support sustainable regional development.When discussing new sustainable projects concerning ecological farming or public service missions in cities, equines could be included in the list of suggested alternatives based on the assets presented in this paper if all economic, social, and welfare conditions are fulfilled.The new 2020 CAP is in process. At a national level, its measures could better support the equine industry through new agri-environmental measures for equines, such as the use of animal traction, the practice of mixed grazing, the use of local threatened breeds, and the use of equines to maintain vacant suburban plots of land.

Finally, by gathering the available scientific knowledge about equine green assets, this paper offers some common reflections and issues about the place of equines in a sustainable regional development.

## 5. Conclusions

The equine industry is constantly evolving according to changes in society. One of the next steps is linked to the growing environmental awareness. This issue concerns citizens but also the political spheres, thereby putting pressure on the stakeholders of all economic sectors, including the equine industry. Indeed, in most European countries, environmental issues are not yet considered to be important enough by stakeholders in the equine industry. However, through their green assets, equines can have an active role in ecological transition and debates, both alone and as a complement to other economic productions and services. In the future, it could be interesting to support knowledge exchange in order to progress equine research, thus making this industry more visible and understandable, and to include equines in political debates about the environment and raise awareness about equine uses to avoid radical actions from animal activists. Creating and publishing all kinds of communication media, such as articles, photos, videos, websites, and podcasts, could be a way to reach a larger audience and make equine owners adapt their management practices to better use equine green assets.

From European organizations to society, everyone should be aware of the potential place of equines during the ecological and agronomic transition toward a greener future.

## Figures and Tables

**Figure 1 animals-10-00106-f001:**
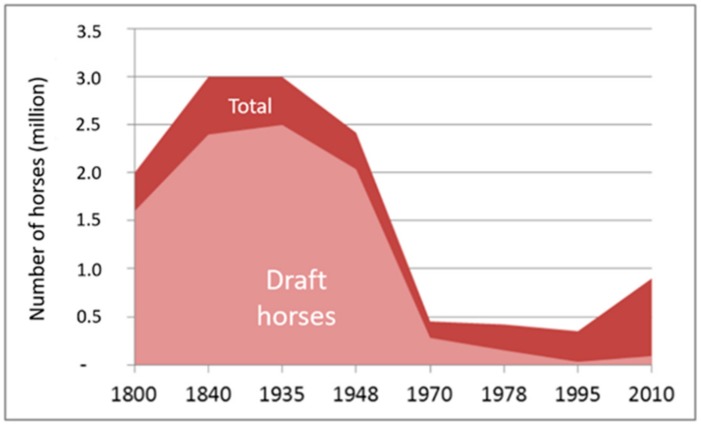
Evolution of the horse population in France from 1800 to 2010 (translated from French [8]).

**Figure 2 animals-10-00106-f002:**
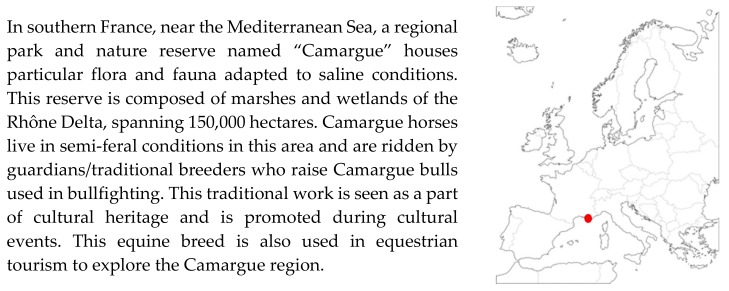
Presentation of the Camargue region and the use of the local equine breeds in this area.

**Figure 3 animals-10-00106-f003:**
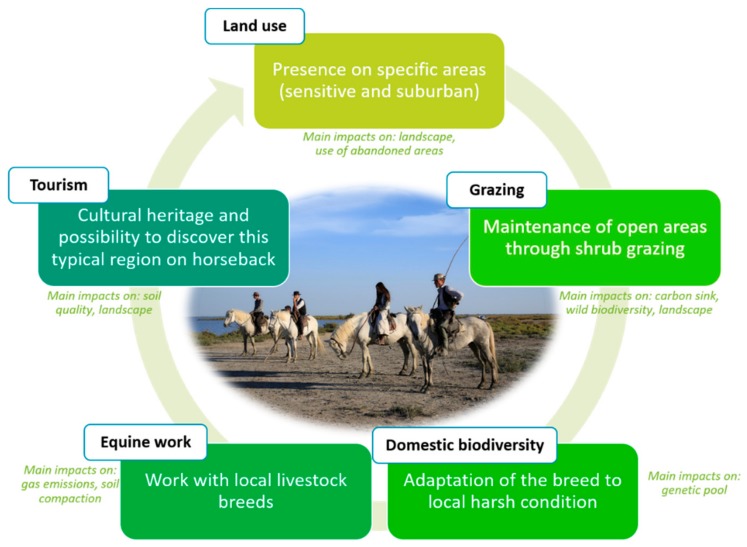
The Camargue horse breed as an image of the link between the five green assets.

**Table 1 animals-10-00106-t001:** Distribution of interviewees according to the scale of expertise and the function.

Scale of Expertise/Function *	Researchers	Institutional Stakeholders	Professional Representatives
France	3	3	3
Europe	1	4	1

* Respondents who answered questions about the French equine industry are counted in the group “Scale of expertise: France.” Those who answered questions with a European point of view are counted in “Scale of expertise: Europe”.

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
