# Peer review of "Green Assets of Equines in the European Context of the Ecological Transition of Agriculture"

_animals, 2020, doi:10.3390/ani10010106_

Round 1
Reviewer 1 Report
This manuscript is very difficult to follow. The English needs to be improved but in also lacks consistency in reasoning. Some examples:
In the introduction the authors seem to blame the EU (48) for the reduction in equine numbers, ignoring the market situation and the lack of demand.
It says a ‘multifunctional approach was chosen’ (65. What does this mean?
A ‘diagnosis based on sectors…. has been carried out’ (66). Where, how?
The ‘green assets have been classified in five groups’ (69). How, on what basis? Should this categorization not be a result of the review?
It says these five domains have unique environmental impact (83). In what sense unique?
No information is provided on how the literature search for this review was done (which data bases, search terms, etc.).
I don’t feel that the conclusion in lines 1711-172 covers the foregoing text. Similar for 191—194.
The text 428-435 does not refer to the definition in 386-387.
Reviewer 2 Report
Dear Authors,
First of all, I would like to pinpoint that there is a lack of papers regarding the use of equine in ecological agriculture. I agree that the time has come and the European equine community should rise and highlight the importance of Equines in sustainable development. Furthermore, horses are undoubtedly part of European culture and all communities should work towards to find them a relevant place in progressive urbanization and agriculture mechanization. The presented review indeed identifies "green assets" and overall is well written. I truly appreciate the huge effort made for finding and summarise the information provided. The manuscript is very valuable with a strong French background and probably, for this reason, contains and evaluates information only from one point of view with some addiction of examples from Poland.
The horse as livestock plays a prominent role in the production and thus has an impact on the environment not only by land use, grazing, tourism, and work but is also a source of meat and milk.
Scientific reliability does not allow to omit also difficult topics. In the whole review the all sector of meat and milk has been missed. It is known that different communities and countries treat horses more like a pet or more like livestock, nevertheless, in a reliable review, especially when raising agriculture-related topics in the context of the whole of Europe this branch of agriculture can not be omitted under any circumstances.
specific comments:
ln 48: are you definitely sure that horses were ignored? please add a citation.
ln 55: what authors mean by legal status?
ln 56: All equines in Europe are counted due to the regulations and identification procedures.
ln 65-71: and again, if you writing in European context do not omit horse as a meat source.
ln 79: what authors exactly mean by the production system?
ln 84-96: consequently - equines as a source of food.
ln 325: This is not natural selection. Since domestication humans create diverse breeds. please rewrite the sequence and add the proper citation.
ln 331: I am not quite sure what authors intend to say about "difficult to define ...breed". Please rewrite and add the proper citations.
ln 347-349: This sentence is not true. I would say not only Arabians but also Tb's were used to creates and shapes the breeds. They were rather used not for consanguinity and population fall but to improve traits of interest. Please rewrite.
ln 350-351: Not at all. Consequently, the high specialization of modern breeds is actually resulting from an intensive directed selection. Please rewrite sentences for better flow and ad proper citation. Małopolska horses are not good general example of this topic.
ln 352: homogenization??? or homozygosity?
ln 553: In Poland equine sector is defined by Arabians, heavy horses used mainly us meat source, and then native ponies for leisure and tourism. Please correct the reliability of the information.
Reviewer 3 Report
Dear Authors, this paper is highlighting the potential of horses or horse keeping as green assets for sustainable development of our society. It is a very important subject and I believe it is of high interest to increase awareness of this on several structural layers in the equine industry. However, it is also important that the information given in e.g. scientific papers within this area is well structured and clearly written, and there is where my concerns are with this specific manuscript. I think the paper could be suitable for publication after revising it, including language editing, but it will need major revision.
Overall, the paper needs to address problems, solutions and knowledge gaps much more clearly, for each of the sections. Readability would be improved if repetitions are avoided.
I have provided more specific comments to the authors, which I hope will be of some help in the revision.
L 29: green alternative to what?
L 32 and elsewhere: the term ”ecological transition” is used quite often but has not been explained. Ecological transition of agriculture could mean many different things.
Fig 1: Horses other than draft horses – where they horses used by the army? Or civilian horses? Or both?
L 48: were horses completely ignored everywhere?
L 56: is that really the case in all member states ? Also, evene if a large proportion of horses are not owned by farmers, they may be housed at a farmers place and then still ba counted.
L58: which differences ? needs to be more specific
L61: revise sentence starrting with However. The will of whom?
L 64: superscript for ”1”
L 72-73: no apology needed for this, but it is wise to explain why. However, some new papers are also available, e.g. Ringmark et al. 2019, Animals 9, 500 or Garrido et al., 2019, J Appl Ecol 56, 946-955. In addition, You are in many places using references in French language where there are references available in English. This could and should be revised.
L 80-99: it is not clear where or how these five assets were selected.
Line 99: this way of writing, ending a sentence with … is in my opinion not acceptable in a published scientific paper. This applies to all places in the manuscript where this occurs, however I have not marked them all in these comments.
Line 104-105: I don’t get it: the Swedish data shows that 75 % of all horses are present in the most inhabited areas in that country. How does that show that horses spread on all kinds of areas?
L 101: revise sentence. Equines require grazing areas, not their feeds. Production of forage (hay, silage and so on) requires areas for grass production. I guess language issues are present here.
Line 110-111: a general rule in writing is to not refer to headings (e.g. when you start a new paragraph). In Line 111 you write ”these areas” and thereby refer to the heading in L 110, but when written correclty it should say: ”Equines are kept in suburban areas thanks to the urban….”. This applies to all places in the manuscript where this writing practice has been used, however I have not marked them all in these comments.
L 119: if you mean feed production I get it, but not if you mean food production.
L120-125: needs to be clarified with clear aim of what you want to highlight.
L127: nature not Nature
L138: the term reject is not very appropriate to use in this context. Manure, faeces, droppings etc could be used instead.
L151: slope not slop
L152: is it specifically animal breeding? Not animal husbandry in general?
L 163: it does not always lead to maintenance of open areas but it may lead to.
L 183: Unclear what yoy mean with ”They”
L 198: not only proteins. The whole plant has a higher digestibility when it is young. Plenty of references in English are available on this matter.
L 203-204: the sentence needs to be revised. As it stands, you are comparing the rumen of the ruminant and the equine but the equine has no rumen.
L 208-209: language issues – horses can not ”be adapted” to low starch diets because they are already designed for low starch diet.
L 211: replace ”In fact” with ”However” to get it right.
L 216: Iceland not Island
L 220-221: last sentence is not clear, rewrite.
L 232: not clear what you refer to by ”It”?
L 234: this sentence needs to have correct grammar.
L 239-243: repetitive. Please merge with previous text, avoid repetitions.
L 247: ”because they are…”
L250: ”development of…”
L 252: How?
L 265: which, and how?
L 268-272: not clear, you are taling about soem results as if the reader knows these results and studies already. Perhaps turning the paragraph ”upside-down” could make the text more clear.
L 281: poor nutritive value of what?
L 335: choice of term for ”boast” ?
L 345-359: This paragraph is unecessary long and provide too much detail for this papers scope.
L 361-362. It is not clear how horses are used in catte breeding in the Camargue area. Remember that the journal is international. This is especially important as most of your references are national and in French meaning that they are not available for all readers.
L 363: Henson? please explain.
L 370-379: This paragraph seems to not have been finsished. Please clarify how this realtes to the aim stated in the paper.
395-398: same comment as for L 361-3621, and L363.
L 438-439: sorry but I don’t get this senetence.
L 441: I am not so sure about sport and race horses, as several brands ar now manufacturing mechanical horses with vrtual reality goggles for practice.
L 443: Is skidding an actual word used in this context?
L 477: avoid use of words like ”better”, but use proper term to describe the actual situation instead.
L 479: it is quite difficult to evaluate how realistic these figures are from the very brief description of the model used.
L 569: other ways to use manure are also available, recycling and fast-composting to soil used in the city parks and so on.
L 577: where has it been reported that native horse breeds needs less medicines ??
L 601: nutritional and welfare requirements of equines are equally important in this context.
L 634-645: I think these lines should be omitted from this paper, save it for another paper. There is no way for the reader to evaluate this information as there is no description of how the interviews were performed, methods for handling and interpreting data and so on.
L 637: yes, exactly my point in this feedback. It is a very important step, but it needs to be highlighted earlier in the paper that this data is lacking.
L 653-658: probably very true but can this be concluded from the current literature study? I would say no.
Round 2
Reviewer 2 Report
Dear Authors,
Thank you for improving the manuscript. Through the reading, I don’t have the impression that meat production has been omitted. I find few concerns but after clarifying those, the manuscript is suitable for publishing in Animals.
Ln 43-44. Is this sentence refers only to heavy horses or horses at all? Please clarify sentence
Ln 211-212 hm. This sentence is awkward. Racehorses need high-quality feed
Ln 219: hardy breeds? I suggest precise e.g primitive?
Ln 221 cattle?
Ln 246-248. This point needs further explanation. I assume you are describing free-living herd of horses and interactions with other populations as wolfes limiting equines. Please rewrite for better flow.
Ln 254-255: move "in Sweden" on the first in the sentence
Ln 267: place citation at the end of the sentence
Ln 272: This sentence is too general.
Ln 273: risks of what?
Author Response
Dear Reviewer,
Thank you a lot for your comments. We tried our best to follow your suggestions.
Ln 43-44. Is this sentence refers only to heavy horses or horses at all? Please clarify sentence
The sentence was changed to: Before 1950, horses were largely used for agriculture, transportation and the army. It was particularly the case for heavy (or draft horses) (represented in light on figure 1) but also for saddle horses (represented in dark on figure 1).
Ln 211-212 hm. This sentence is awkward. Racehorses need high-quality feed
The sentence was changed to: Most equines (except high performance horses like racehorses) do not need high quality feed [...].
Ln 219: hardy breeds? I suggest precise e.g primitive? Ln 221 cattle?
Indeed, your suggestion seems more accurate. The sentence was changed to: Some primitive equine breeds better mobilize the bodily reserves they gain during summer to survive during the winter if grass is not sufficient [16] than some specialised equine breeds. For example, in Iceland, [...].
Ln 246-248. This point needs further explanation. I assume you are describing free-living herd of horses and interactions with other populations as wolfes limiting equines. Please rewrite for better flow.
We rewritten the paragraph:
The wolf population in Galicia, Spain, was maintained partly thanks to ponies bred in semi-natural conditions. These once were the preferential feed source of wolves [28] which consequently did not attack other livestock. As a result, farmers felt less disposed to shoot them down. Finally, ponies were an indirect way to conserve the wolf population.Ln 254-255: move "in Sweden" on the first in the sentence
Changes done.
Ln 267: place citation at the end of the sentence
Changes done.
Ln 272: This sentence is too general. Ln 273: risks of what?
We changed the sentence: Moreover, in winter, equine can remain on pastures and continue maintaining them while cattle are held in stables [16]. However, mixing cattle and equine grazing can generate problems, especially when increasing the number of animals in the pasture. For example, when the equine stocking rate is too high, [...].